# A Deep Learning Model for Cervical Cancer Screening on Liquid-Based Cytology Specimens in Whole Slide Images

**DOI:** 10.3390/cancers14051159

**Published:** 2022-02-24

**Authors:** Fahdi Kanavati, Naoki Hirose, Takahiro Ishii, Ayaka Fukuda, Shin Ichihara, Masayuki Tsuneki

**Affiliations:** 1Medmain Research, Medmain Inc., Fukuoka 810-0042, Fukuoka, Japan; fkanavati@medmain.com; 2Department of Clinical Laboratory, Sapporo Kosei General Hospital, 8-5 Kita-3-jo Higashi, Chuo-ku, Sapporo 060-0033, Hokkaido, Japan; okh@hotmail.co.jp (N.H.); 141takahiro8260@gmail.com (T.I.); ayaka3843@gmail.com (A.F.); 3Department of Surgical Pathology, Sapporo Kosei General Hospital, 8-5 Kita-3-jo Higashi, Chuo-ku, Sapporo 060-0033, Hokkaido, Japan; trampolineboy@mbn.nifty.com

**Keywords:** liquid-based cytology, deep learning, cervical screening, whole slide image

## Abstract

**Simple Summary:**

In this pilot study, we aimed to investigate the use of deep learning for the classification of whole-slide images of liquid-based cytology specimens into neoplastic and non-neoplastic. To do so, we used a large training and test sets. Overall, the model achieved good classification performance in classifying whole-slide images, demonstrating the promising potential use of such models for aiding the screening processes for cervical cancer.

**Abstract:**

Liquid-based cytology (LBC) for cervical cancer screening is now more common than the conventional smears, which when digitised from glass slides into whole-slide images (WSIs), opens up the possibility of artificial intelligence (AI)-based automated image analysis. Since conventional screening processes by cytoscreeners and cytopathologists using microscopes is limited in terms of human resources, it is important to develop new computational techniques that can automatically and rapidly diagnose a large amount of specimens without delay, which would be of great benefit for clinical laboratories and hospitals. The goal of this study was to investigate the use of a deep learning model for the classification of WSIs of LBC specimens into neoplastic and non-neoplastic. To do so, we used a dataset of 1605 cervical WSIs. We evaluated the model on three test sets with a combined total of 1468 WSIs, achieving ROC AUCs for WSI diagnosis in the range of 0.89–0.96, demonstrating the promising potential use of such models for aiding screening processes.

## 1. Introduction

According to the Global Cancer Statistics 2020 [1], cervical cancer is the fourth leading cause of cancer death in women, with an estimated 342,000 deaths worldwide in 2020. However, incidence and mortality rates have declined over the past few decades due to either increasing average socioeconomic levels or a diminishing risk of persistent infection with high risk human papillomavirus (HPV) [1]. In developed countries, cervical cytology screening systems have been organised to reduce mortality from cervical cancer [2,3,4,5,6,7,8,9].

The introduction of cervical cancer screening led to a fall in associated mortality rates; however, there is some evidence that the conventional smear method for screening is not consistent in reliably detecting cervical intraepithelial neoplasia (CIN) [10,11,12]. This is because conventional cervical smears, when spread on glass slides, tend to have the cells of interest mixed with blood, debris, and exudate. A number of new technologies and procedures are becoming available in various screening programs (e.g., liquid-based cytology (LBC), automated screening devices, computer-assisted microscopy, digital colposcopy with automated image analysis, HPV testing). The LBC technique preserves the cells of interest in a liquid medium and removes most of the debris, blood, and exudate either by filtering or density gradient centrifugation. The other advantages in LBC are the availability of residual material for HPV and other molecular tests and the connection with automated screening devices. ThinPrep (Hologic, Inc., Marlborough, MA, USA) and SurePath (Becton Dickinson, Inc., Franklin Lakes, NJ, USA) for LBC specimen preparation have been approved by the US Food and Drug Administration (FDA), and it has also been adopted by the cervical screening programme in the UK. Moreover, the ThinPrep collection vial has been approved by the FDA for direct testing for HPV, which is particularly useful for managing women whose Pap smear tests show atypical squamous cells (ASCs) [4,13].

In 1998, the FDA approved the FocalPoint Slide Profiler (Becton Dickinson, Inc.) as a primary automated screener for cervical smears, followed by approval in 2002 for use with SurePath slides. In 2003, the FDA approved the ThinPrep Imaging System (Hologic, Inc.) as a primary screener for ThinPrep Pap slides. The FocalPoint uses algorithms to measure cellular features (e.g., nuclear size, integrated optical density, nuclear to cytoplasmic ratio, and nuclear contour) for the diagnosis of squamous and glandular lesions [14]. In the US, the American Society of Cytopathology (ASC) established guidelines for automated Pap test screening using the ThinPrep Imaging System and the FocalPoint GS Imaging System [15]. However, there are some issues with the current automated screening support systems. A multi-institutional feasibility study in Japan validated the usefulness of FocalPoint for cervical cytology automated screening quality control and showed that it was useful for NILM (Negative for Intraepithelial Lesion or Malignancy) cases, but on the other hand, 2174 (18.1%) of 12,000 specimens were judged to be unmeasurable and were not evaluated [16]. In the US, unmeasured rates were reported to be as low as 2.5% [17], 5.9% [18], and 4.8% [19], while in Brazil, the unmeasured rate was very high at 30.8% [20]. In order to use FocalPoint, it was reported that the unmeasured ratio can be suppressed to a low value by adjusting a specimen preparation method(s) including staining [16]. However, in routine clinical practice, there are many screening facilities that do not (or cannot) stain specimens accordingly to adjust for FocalPoint, as reported in Japan and Brazil [16,20].

The sensitivity of conventional cytology cervical cancer screening for detecting pre-invasive squamous and glandular lesions (pre-invasive intraepithelial lesions) is clearly far from perfect. It has been reported that most studies of the conventional Pap test were severely biased, and it was only moderately accurate and did not achieve concurrently high sensitivity and specificity (i.e., sensitivity ranged from 30% to 87% and specificity ranged from 86% to 100%) [21]. Moreover, the sensitivity of conventional cervical cytology is less than ideal for invasive cancers, with a wide range (45% to 76%), and false-negative or false-unsatisfactory rate in conventional smears was 50% [22]. These studies indicate that many women with cervical cancer have a history of one or more negative cervical cytology reports. As a background of these results, the interobserver reproducibility of cervical cytology is less than perfect. The reproducibility of 4948 monolayer cytologic interpretations was moderate (kappa = 0.46; 95% confidence interval (CI), 0.44–0.48) among four categories of diagnosis (i.e., negative, ASC-US, LSIL, and over HSIL) by multiple well-trained observers [23]. In the same study, the greatest disagreement in monolayer cervical cytology involved ASC-US interpretations. Of the 1473 original interpretations of ASC-US, the second reviewer concurred in only 43.0% [23].

Whole-slide images (WSIs) are digitisations of the conventional glass slides obtained via specialised scanning devices (WSI scanners), and they are considered to be comparable to microscopy for primary diagnosis [24]. A routine scanning of LBC slides in a single layer of WSIs would be suitable for further high throughput analysis (e.g., automated image based cytological screening and medical image analysis) [25]. The advent of WSIs led to the application of medical image analysis techniques, machine learning, and deep learning techniques for aiding pathologists in inspecting WSIs. Deep-learning-based applications ranged from tasks, such as cancer diagnosis from WSIs, cell classification, and segmentation of nuclei, to patient stratification and outcome prediction [26,27,28,29,30,31,32,33,34,35,36,37,38,39,40,41,42,43,44]. For cytology, in particular, only recently have there been investigations for applying deep learning on large datasets of cervical WSIs Holmström et al. [45], Lin et al. [46], Cheng et al. [47].

In this pilot study, we trained a deep learning model, based on convolutional and recurrent neural networks, using a dataset of 1605 cervical WSIs. We evaluated the model on three test sets with a combined total of 1468 WSIs, achieving ROC AUCs for WSI diagnosis in the range of 0.89–0.96.

## 2. Materials and Methods

### 2.1. Clinical Cases and Cytopathological Records

This is a retrospective study. A total of 3121 LBC ThinPrep Pap test (Hologic, Inc.) conventionally prepared cytopathological slide glass specimens of human cervical cytology were collected from a private clinical laboratory in Japan after cytopathological review of those specimens by cytoscreeners and pathologists. The cases were selected mostly at random so as to reflect a real clinical scenario as much as possible; we have also collected cases so as to compile a test set with an equal balance of neoplastic and NILM. The cytoscreeners and pathologists excluded cases that had poor scanned quality (n=32). Each WSI diagnosis was observed by at least two cytoscreeners and pathologists, with the final checking and verification performed by a senior cytoscreener or pathologist. All WSIs were scanned at a magnification of ×20 using the same Aperio AT2 digital whole-slide scanner (Leica Microsystems, Osaka, Japan) and were saved in SVS file format with JPEG2000 compression.

### 2.2. Dataset

Table 1 breaks down the distribution of the dataset into training, validation, and test sets. The split was carried out randomly taking into account the proportion of each label in the dataset. A clinical laboratory that provided LBC cases was anonymised. The test sets were composed of WSIs of full agreement, clinical balance, and equal balance LBC specimens. The full agreement test set consisted of NILM and neoplastic LBC cases whose obtained diagnoses were fully agreed by two independent cytoscreeners in different institutes. The clinical balance test set consisted of 95% NILM and 5% neoplastic LBC cases based on a real clinical setting [48,49]. The equal balance test set consisted of 50% NILM and 50% neoplastic LBC cases. NILM and neoplastic LBC cases for clinical and equal balance test sets were collected based on the diagnoses provided by the clinical laboratory. The cases in the clinical and equal balance test sets were only based on the diagnostic reports. From these two test sets, we have also created their reviewed counterparts (clinical balance reviewed and equal balance reviewed), where two independent cytoscreeners viewed all the cases and the ones they had a disagreement on were removed (see Table 1).

### 2.3. Annotation

Senior cytoscreeners and pathologists who perform routine cytopathological screening and diagnoses in general hospitals and clinical laboratories in Japan manually annotated 352 neoplastic WSIs from the training sets. Coarse annotations were obtained by free-hand drawing. (Figure 1 using an in-house online tool developed by customising the open-source OpenSeadragon tool at https://openseadragon.github.io/ (accessed on 10 January 2020), which is a web-based viewer for zoomable images.) On average, the cytoscreeneers and pathologists annotated 150 cells (or cellular clusters) per WSI.

Neoplastic WSIs consisted of ASC (atypical squamous cell), LSIL (low-grade squamous intraepithelial lesion), HSIL (high-grade squamous intraepithelial lesion), CIS (carcinoma in situ), ADC (adenocarcinoma), and SCC (squamous cell carcinoma), except for the NILM. For example, on the HSIL (Figure 1A–D) and SCC (Figure 1E–H) WSIs, cytoscreeners and pathologists performed annotations around the neoplastic cells (Figure 1B–D,F–H) based on the representative neoplastic epithelial cell morphology (e.g., increased nuclear/cytoplasmic ratio, abnormalities of nuclear shape, hyperchromatism, irregular chromatin distribution, and prominent nucleolus). On the other hand, the cytoscreeners and pathologists did not annotate areas where it was difficult to cytologically determine that the cells were neoplastic. The NILM subset of the training and validation sets (1301 WSIs) was not annotated and the entire cell spreading areas within the WSIs were used.

The average annotation time per WSI was about an hour. Annotations performed by the cytoscreeners and pathologists were modified (if necessary), confirmed, and verified by a senior cytoscreener.

### 2.4. Deep Learning Models

Our deep learning models consisted of a convolutional neural network (CNN) and a recurrent neural network (RNN) that were trained simultaneously end to end. For the CNN, we have used the EfficientNetB0 architecture [50] with a modified input size of 1024 × 1024 px to allow a larger view; this is based on cytologists’ input that they usually need to view the neighbouring cells around a given cell in order to diagnose more accurately. We then performed 7 × 7 max pooling with a stride of 5 × 5. The output of the CNN was reshaped and provided as input to an RNN with a gated recurrent unit Cho et al. [51] model of size 128, followed by a fully connected layer. We used the partial fine-tuning approach [52] for the tuning the CNN component, where only the affine weights of the batch normalisation layers are updated while the rest of the weights in the CNN remain frozen. We used the pre-trained weights from ImageNet as starting weights. Figure 2 shows a simplified overview of the model. The RNN component was initialised with random weights.

WSIs tend to contain a large white background that is not relevant for the model. We therefore start the preprocessing by eliminating the white background using Otsu’s method [53] applied to the greyscale version of the WSIs.

For training and inference, we then proceeded by extracting 1024 × 1024 px tiles from the tissue regions. We performed the extraction in real-time using the OpenSlide library [54]. To perform inference on a WSI, we used a sliding window approach with a fixed-size stride of 512 × 512 px (half the tile size). This results in a grid-like output of predictions on all areas that contained cells, which then allowed us to visualise the prediction as a heatmap of probabilities that we can directly superimpose on top of the WSI. Each tile had a probability of being neoplastic; to obtain a single probability that is representative of the WSI, we computed the maximum probability from all the tiles.

During training, we maintained an equal balance of positively and negatively labelled tiles in the training batch. To do so, for the positive tiles, we extracted them randomly from the annotated regions of neoplastic WSIs, such that within the 1024 × 1024 px, at least one annotated cell was visible anywhere inside the tile. For the negative tiles, we extracted them randomly anywhere from the tissue regions of NILM WSIs. We then interleaved the positive and negative tiles to construct an equally balanced batch that was then fed as input to the CNN. In addition, to reduce the number of false positives, given the large size of the WSIs, we performed a hard mining of tiles, whereby at the end of each epoch, we performed full sliding window inference on all the NILM WSIs in order to adjust the random sampling probability such that false positively predicted tiles of NILM were more likely to be sampled.

During training, we performed real-time augmentation of the extracted tiles using variations of brightness, saturation, and contrast. We trained the model using the Adam optimisation algorithm [55], with the binary cross entropy loss, $beta_{1}=0.9$, $beta_{2}=0.999$, and a learning rate of 0.001. We applied a learning rate decay of 0.95 every 2 epochs. We used early stopping by tracking the performance of the model on a validation set, and training was stopped automatically when there was no further improvement on the validation loss for 10 epochs. The model with the lowest validation loss was chosen as the final model.

### 2.5. Interobserver Concordance Study

For the interobserver concordance study, a total of 10 WSIs (8 NILM cases and 2 neoplastic cases) of cervical LBC already reported by a clinical laboratory were retrieved from the records. Using the in-house on-line web virtual slide application, a total of 16 cytoscreeners (8 have over 10 years experiences and 8 have less than 10 years experiences) have reviewed the 10 WSIs and reported in subclasses (NILM, ASC-US, ASC-H, LSIL, HSIL, SCC, ADC).

### 2.6. Software and Statistical Analysis

The deep learning models were implemented and trained using the open-source TensorFlow library [56].

To assess the cytopathological diagnostic concordance of cytoscreeners, we performed the Fleiss’ kappa statistic, which is a measure of inter-rater agreement of a categorical variable [57] between two or more raters. We calculated the kappa values using Microsoft Excel 2016 MSO (16.0.13029.20232) 64 bit. The scale for interpretation is as follows: ≤0.0, poor agreement; 0.01–0.20, slight agreement; 0.21–0.40, fair agreement; 0.41–0.60, moderate agreement; 0.61–0.80, substantial agreement; 0.81–1.00, almost perfect agreement. AUCs were calculated in python using the scikit-learn package [58] and plotted using matplotlib [59]. The 95% CIs of the AUCs were estimated using the bootstrap method [60] with 1000 iterations.

The true positive rate (TPR) was computed as
(1)$$TPR=\frac{TP}{TP+FN}$$
and the false positive rate (FPR) was computed as
(2)$$FPR=\frac{FP}{FP+TN}$$
where TP, FP, and TN represent true positive, false positive, and true negative, respectively. The ROC curve was computed by varying the probability threshold from 0.0 to 1.0 and computing both the TPR and FPR at the given threshold.

### 2.7. Code Availability

We adapted the training code from https://github.com/tensorflow/models/tree/master/official/vision/image_classification (accessed on 14 February 2020).

## 3. Results

### 3.1. High AUC Performance of WSI Evaluation of Neoplastic Cervical Liquid-Based Cytology (LBC) Images

The aim of this retrospective study was to train a deep learning model for the classification of neoplastic cervical WSIs. We trained a model that consists of a convolutional and a recurrent neural network using a dataset of 1503 WSIs for training and 150 for validation. We evaluated the model on three test sets with a combined total of 1468 WSIs. Figure 3 shows the resulting ROC curves, and Table 2 lists the resulting ROC AUC and log loss, as well as the accuracy, sensitivity, and specificity computed at a probability threshold of 0.5. Table 3 shows the confusion matrix. The model achieved a good performance overall, with ROC AUCs of 0.96 (0.92–0.99) on the full agreement, 0.89 (0.81–0.96) on the clinical balance reviewed, and 0.92 (0.89–0.94) on the equal balance reviewed test sets.

### 3.2. True Positive Prediction

Our deep learning model satisfactorily predicted neoplastic epithelial cells (Figure 4C–G) in cervical LBC (Figure 4A,B) specimen. The heatmap image shows true positive predictions (Figure 4B–D) of neoplastic epithelial cells. In contrast, in low probability tiles (Figure 4H,I), two independent cytoscreeners confirmed there were no neoplastic epithelial cells.

### 3.3. True Negative Prediction

Our model satisfactorily predicted NILM cases (Figure 5A,B) in cerevical LBC specimen. The heatmap image shows true negative predictions (Figure 5B,D,E) of neoplastic epithelial cells. In both zero (Figure 5C) and very low probability tiles (Figure 5D,E), there are no neoplastic epithelial cells.

### 3.4. False Positive Prediction

A cytopathologically diagnosed NILM case (Figure 6A) was false positively predicted for neoplastic epithelial cells (Figure 6B). The heatmap image (Figure 6B) shows false positive predictions of neoplastic epithelial cells (Figure 6C,E) with high probabilities. Cytopathologically, there are parabasal cells with a high nuclear cytoplasmic (N/C) ratio (Figure 6C,D) and cell clusters of squamous epithelial cells with cervical gland cells with high N/C ratios (Figure 6E), which could be a major cause of false positive.

### 3.5. Interobserver Variability

To evaluate the practical interobserver variability among cytoscreeners, we have asked a total of 16 cytoscreeners (8 are over 10 years experiences and 8 are less than 10 years experiences) to review the same 10 LBC WSIs, which consist of 8 NILM and 2 neoplastic cases already diagnosed by a clinical laboratory. The results of each cytoscreener were summarised in Table 4. The Fleiss’ kappa statistics were summarised in Table 5. There was poor to moderate concordance in assessing subclass, with Fleiss’ kappas of NILM (range: 0.042–0.755), neoplastic (range: 0.098–0.500), and all cases (range: 0.364–0.716). On the other hand, there was poorly to almost perfect concordance in assessing binary class, with Fleiss’ kappas of NILM (range: 0.073–0.815), neoplastic (1.000), and all cases (range: 0.568–0.861). Interestingly, there was a robust higher concordance in both subclass and binary class among cytoscreeners over 10-year experiences. However, overall, there was poor concordance in assessing NILM cases (range: 0.042–0.073).

## 4. Discussion

In this pilot study, we trained a deep learning model for the classification of neoplastic cells in WSIs of LBC specimens. The model achieved overall a good performance, with ROC AUCs of 0.96 (0.92–0.99) on the full agreement, 0.89 (0.81–0.96) on the clinical balance reviewed, and 0.92 (0.89–0.94) on the equal balance reviewed test sets.

Looking at the interobserver concordance among cytoscreeners in Table 4, it is obvious that there is considerable interobserver variability, with the poor concordance in NILM cases even for binary classification (NILM vs. neoplastic). In addition, there is the problem of human fatigue due to the continuous observation of a large number of cases. Therefore, when considering future accuracy control, it may be necessary to conduct screening using deep learning model(s) with guaranteed accuracy, such as the results of this study, at least in the binary classification (NILM vs. neoplastic), and to conduct detailed assessments by cytoscreeners and cytopathologists in the subclassification (e.g., NILM, ASC-US, ASC-H, LSIL, HSIL, SCC, and ADC).

From our results in Figure 2, it was obvious that there was interobserver variability among cytoscreeners in different clinical laboratories and hospitals. Clinical balance and equal balance test sets were prepared based on diagnostic (screening) reports from a clinical laboratory. The only difference between clinical balance and clinical balance-reviewed (same as equal balance and equal balance-reviewed) was whether it was additionally reviewed by two more cytoscreeners in different clinical laboratories and hospitals or not. All scores (ROC-AUC, accuracy, sensitivity, and specificity) were increased in clinical balance-reviewed and equal balance-reviewed test sets as compared to clinical balance and equal balance test sets (Figure 2). Hence, our deep learning model would be helpful for standardising in the screening process.

In routine cervical cancer screening at clinical laboratories and hospitals, it is difficult to introduce a screening programme dependent on cervical smears due to poor human cytoscreener resources. LBC techniques opened new possibilities for a systemic cervical cancer screening. LBC slides are amenable to high throughput automated analysis. Especially for the detection of rare events on LBC slides, WSI and subsequent image analysis is of crucial importance for guaranteeing a standardised high-quality read out [25]. Practical automated cervical cytology screening devices have been under development since the 1950s. The technological development in semi-automated screening devices for cervical cancer screening is very rapid; however, currently, no machines are available to provide a fully automated screening by computer without human intervention. There are two FDA-approved semi-automated slide scanning devices on the market; these systems are the BD FocalPoint GS Imaging System and the HOLOGIC ThinPrep Imaging System. Both are designed to perform computer-assisted analysis of cellular images followed by location-guided screening of limited fields of view. FocalPoint-assisted smear reading has been proposed prior to conventional manual reading; the latter may be unnecessary for cases reported as No Further Review (NFR) and would be required for cases reported as Review (REV) [61]. FocalPoint-assisted practice showed statistically superior sensitivity and specificity when compared to conventional manual smear screening for the detection of HSIL and LSIL [14,62,63]. However, ASC-US sensitivity and specificity were not significantly different between FocalPoint-assisted practice and conventional screening [62]. Overall, in neoplastic slides (ASC-US, LSIL, and HSIL) by FocalPoint-assisted practice, sensitivity was in the range of 81.1–86.1% and specificity was in the range of 84.5–95.1% [62]. The other study showed that FocalPoint-assisted reading was comparable to conventional reading, and the very low observed negative predictive value of an NFR report (0.02%) suggested that these cases might safely return to periodic screening [61]. The ThinPrep Imaging System (TIS) is an automated system that uses location-guided screening to assist cytoscreeners in reviewing a ThinPrep Pap LBC slides [64]. TIS scans the LBC slides and identifies 22 fields of view (FOVs) on each slide based on optical density measurements and other features [64]. It has been reported that TIS was ideally suited to the rapid screening of negative cases; however, the sensitivity and specificity of the TIS (85.19% and 96.67%, respectively) were equivalent to those of manual screening (89.38% and 98.42%, respectively) [65]. In another study, for diagnostic categories of neoplastic slides (ASC-US, LSIL, and HSIL) by TIS practice, sensitivity was in the range of 79.2–82.0% and specificity was in the range of 97.8–99.6% [64].

As shown in Figure 2, our LBC cervical cancer screening deep learning model exhibited around 90% accuracy (in the range of 89–91%), 86% sensitivity (in the range of 84–89%), and 91% specificity (in the range of 90–92%) in full agreement, clinical balance-reviewed, and equal balance-reviewed test sets; those scores were as well or better than the existing assistance systems mentioned above.

## 5. Conclusions

In the present study, we have trained a deep learning model for the classification of neoplastic cervical LBC in WSIs. We have evaluated the model on three test sets achieving ROC-AUCs for WSI diagnosis in the range of 0.89–0.96. The main advantage of our deep learning model is that the model can be used to evaluate the cervical LBC at the WSI level. Therefore, our model is able to infer whether the cervical LBC WSI is NILM (non-neoplastic) (Figure 5) or neoplastic (Figure 4). This makes it possible to use a deep learning model such as ours as a tool to aid in the cervical screening process, which could potentially be used to rank the cases by order of priority. After which the cytoscreeners will need to perform full screening and subclassification (e.g., ASC-US, ASC-H, LSIL, HSIL, SCC, ADC) on neoplastic output cases after the primary screening by our deep learning model, which could reduce their working time as the model would have highlighted the potential suspected neoplastic regions, and they would not have to perform an exhaustive search through the entire WSI.

## Figures and Tables

**Figure 1 cancers-14-01159-f001:**
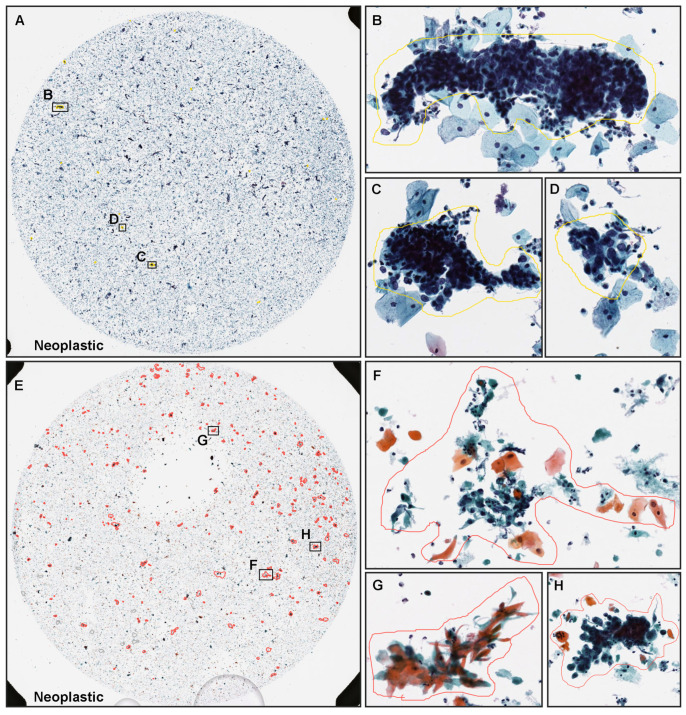
Representative manually drawing annotation images for neoplastic labels on liquid-based cytology (LBC) slides. The LBC case (**A**) was diagnosed as HSIL (high-grade squamous intraepithelial lesion) based on the representative neoplastic squamous epithelial cells with increase in nuclear/cytoplasmic ratio and nuclear atypia (**B**–**D**). The LBC case (**E**) was diagnosed as SCC (squamous cell carcinoma) based on the representative neoplastic squamous epithelial cells with HSIL features (**F**–**H**). Representative neoplastic cells were roughly annotated using in-house on-line drawing tools.

**Figure 2 cancers-14-01159-f002:**
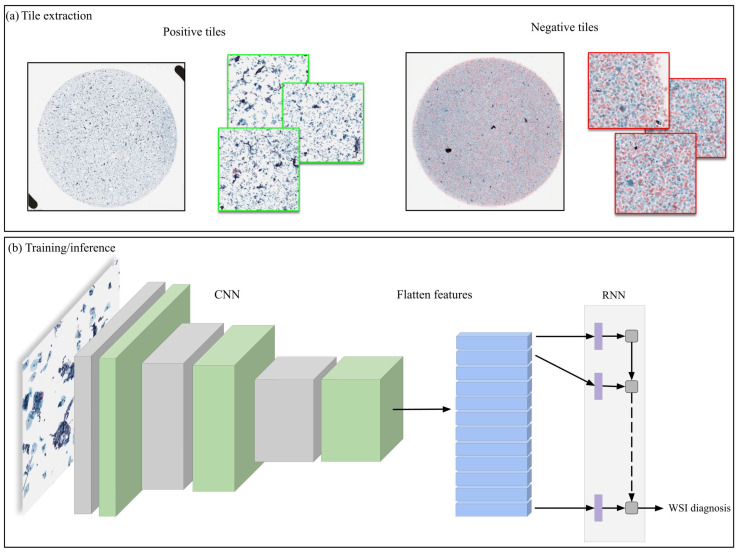
Method overview. (**a**) Large 1024 × 1024 are extracted from the WSIs; for the neoplastic WSIs, tiles are extracted only from annotated regions, while from NILM WSIs, tiles are extracted randomly from any region. (**b**) The tiles are then used to create random balanced batches used to train the model, which is composed of a CNN and an RNN and are trained simultaneously. During inference, the model is applied on all of the tiles of the WSIs in a sliding window fashion, and the WSI label is predicted based on the maximum probability from all of the tiles.

**Figure 3 cancers-14-01159-f003:**
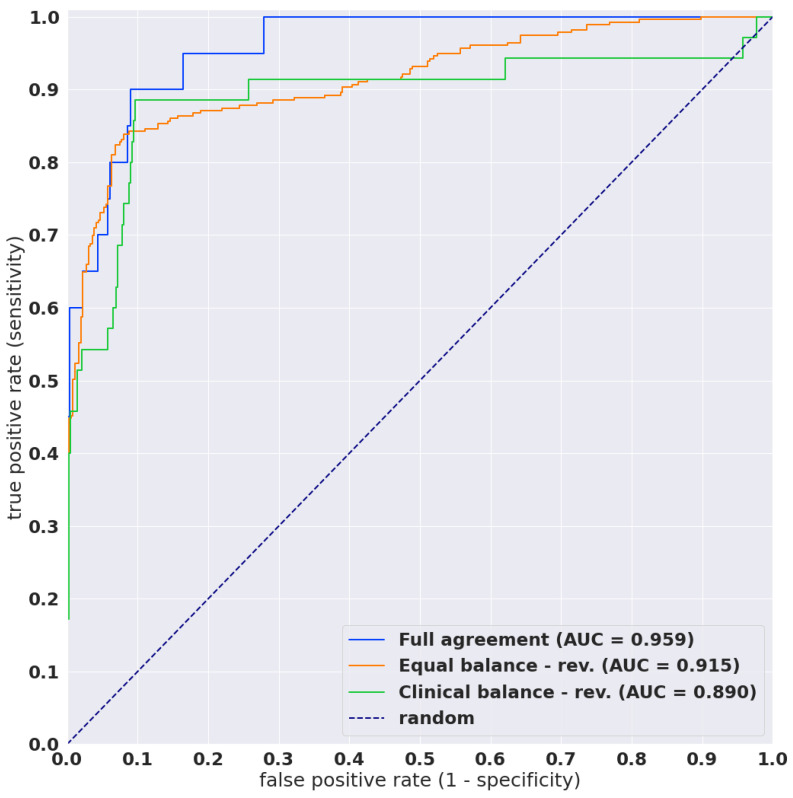
ROC curves for the three test sets.

**Figure 4 cancers-14-01159-f004:**
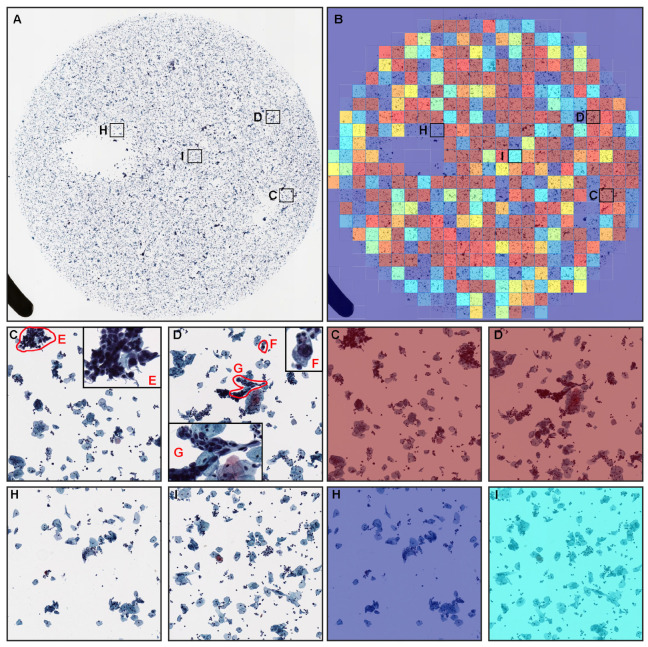
A representative example of neoplastic true positive prediction outputs on a liquid-based cytology (LBC) case from test sets. In the neoplastic whole-slide image (WSI) of LBC specimen (**A**), the heatmap image (**B**) shows a true positive prediction of neoplastic epithelial cells in high probability tiles (**C**,**D**), which correspond, respectively, to neoplastic epithelial cells (**E**–**G**) equivalent to HSIL (high-grade squamous intraepithelial lesion). On the other hand, in low probability tiles (**H**,**I**) of the same heatmap image (**B**), there are no evidence of neoplastic cells.

**Figure 5 cancers-14-01159-f005:**
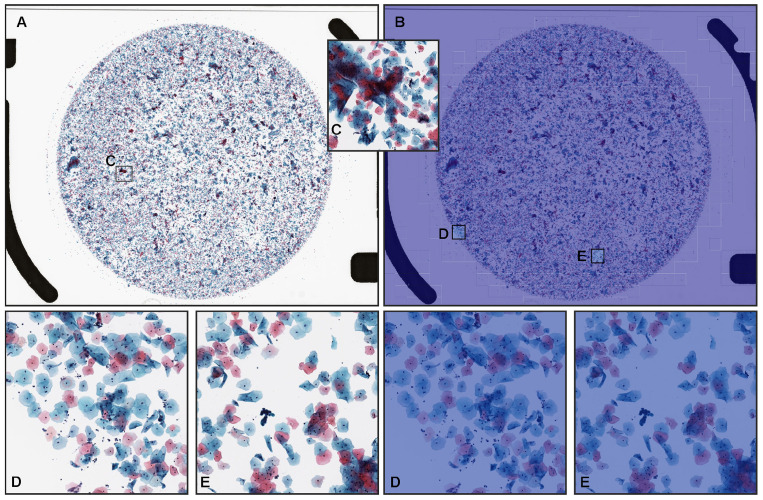
A representative example of neoplastic true negative prediction outputs on a liquid-based cytology (LBC) case from test sets. In the NILM (negative for intraepithelial lesion or malignancy) whole slide image (WSI) of LBC specimen (**A**), the heatmap image (**B**) shows true negative prediction of neoplastic epithelial cells which correspond, respectively, to non-neoplastic epithelial cells (**C**). Moreover, in very low probability tiles (**D**,**E**) of the same heatmap image (**B**), there are no evidence of neoplastic cells.

**Figure 6 cancers-14-01159-f006:**
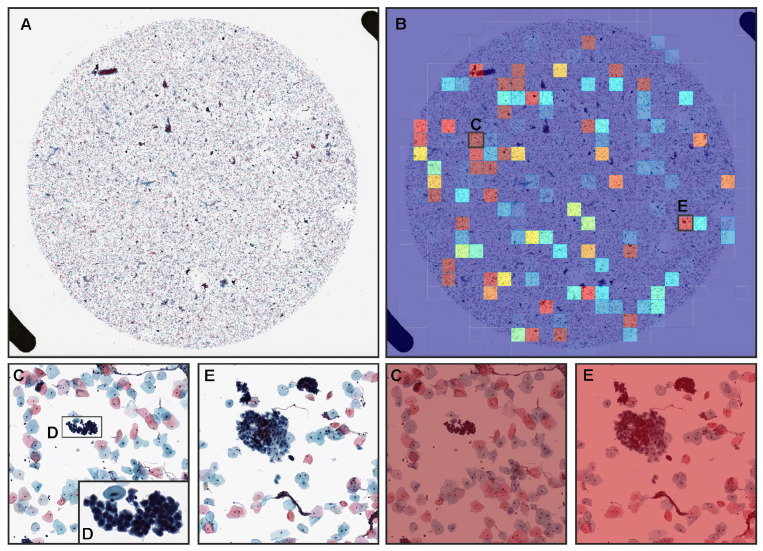
A representative example of neoplastic false positive prediction outputs on a liquid-based cytology (LBC) case from test sets. Cytopathologically, (**A**) is a NILM (negative for intraepithelial lesion or malignancy) whole-slide image (WSI) of LBC specimen. The heatmap image (**B**) exhibited false positive predictions of neoplastic tiles (**C**,**E**). In (**C**), there are parabasal cells with a slightly high nuclear cytoplasmic (N/C) ratio with dense chromatin appearance due to the cellular overlapping (**D**). In (**E**), there are cell clusters of squamous epithelial cells and cervical gland cells with slightly high N/C ratios and a dense chromatin appearance due to the cellular overlapping.

**Table 1 cancers-14-01159-t001:** Distribution of WSIs into training, test, and validation sets.

	Total	Neoplastic	NILM
training	1503	302	1201
validation	150	50	100
test: full agreement	300	20	280
test: equal balance	750	375	375
test: equal balance-rev.	643	279	364
test: clinical balance	750	38	712
test: clinical balance-rev.	525	35	490

**Table 2 cancers-14-01159-t002:** ROC AUC, log loss, accuracy, sensitivity, and specificity results on the test sets.

	Full Agreement	Clinical Balance	Clinical Balance-rev.	Equal Balance	Equal Balance-rev.
ROC AUC	0.960 [0.921–0.988]	0.774 [0.679–0.841]	0.890 [0.808–0.963]	0.827 [0.795–0.852]	0.915 [0.892–0.937]
log loss	2.244 [2.021–2.458]	2.272 [2.141–2.412]	1.347 [1.238–1.465]	1.126 [0.994–1.264]	0.913 [0.794–1.055]
accuracy	0.907 [0.873–0.937]	0.629 [0.591–0.660]	0.903 [0.876–0.924]	0.759 [0.725–0.785]	0.885 [0.859–0.908]
sensitivity	0.850 [0.667–1.000]	0.816 [0.686–0.923]	0.886 [0.774–0.978]	0.624 [0.573–0.668]	0.839 [0.794–0.880]
specificity	0.911 [0.877–0.942]	0.619 [0.579–0.652]	0.904 [0.877–0.926]	0.893 [0.862–0.924]	0.920 [0.890–0.945]

**Table 3 cancers-14-01159-t003:** Confusion matrix.

			Predicted Label	
			NILM	Neoplastic
Full agreement	True label	NILM	255	25
		Neoplastic	3	17
Clinical balance	True label	NILM	441	271
		Neoplastic	7	31
Clinical balance-rev.	True label	NILM	443	47
		Neoplastic	4	31
Equal balance	True label	NILM	335	40
		Neoplastic	141	234
Equal balance-rev.	True label	NILM	335	29
		Neoplastic	45	234

**Table 4 cancers-14-01159-t004:** Cytopathological evaluations for 10 LBC WSIs by diagnostic report (Dx) and 16 cytoscreeners (CS) with their age and years of experience.

Age	Exp. (Years)		Case 1	Case 2	Case 3	Case 4	Case 5	Case 6	Case 7	Case 8	Case 9	Case 10
		Dx	NILM	NILM	NILM	NILM	NILM	NILM	NILM	NILM	HSIL	LSIL
30s	≥10	CS1	NILM	NILM	NILM	NILM	NILM	NILM	NILM	NILM	HSIL	ASC-H
50s		CS2	NILM	NILM	NILM	ASC-H	NILM	NILM	HSIL	ASC-H	HSIL	HSIL
50s		CS3	NILM	NILM	NILM	NILM	NILM	NILM	NILM	ASC-US	HSIL	LSIL
40s		CS4	NILM	NILM	NILM	ASC-US	NILM	NILM	NILM	ASC-US	HSIL	SCC
30s		CS5	NILM	NILM	NILM	NILM	NILM	NILM	NILM	NILM	HSIL	ASC-US
30s		CS6	NILM	ASC-US	NILM	NILM	NILM	NILM	NILM	NILM	HSIL	HSIL
60s		CS7	NILM	NILM	NILM	NILM	NILM	NILM	NILM	NILM	HSIL	ASC-H
40s		CS8	NILM	NILM	NILM	NILM	NILM	NILM	NILM	NILM	HSIL	ASC-US
20s	<10	CS9	NILM	NILM	NILM	NILM	NILM	NILM	NILM	NILM	HSIL	LSIL
20s		CS10	NILM	NILM	NILM	NILM	NILM	NILM	NILM	NILM	LSIL	LSIL
30s		CS11	NILM	NILM	NILM	NILM	ASC-H	NILM	NILM	HSIL	LSIL	HSIL
20s		CS12	NILM	ASC-US	ASC-H	NILM	NILM	NILM	NILM	LSIL	SCC	HSIL
40s		CS13	NILM	NILM	HSIL	NILM	NILM	NILM	NILM	ASC-US	HSIL	ASC-H
30s		CS14	NILM	NILM	LSIL	NILM	NILM	NILM	NILM	NILM	HSIL	LSIL
20s		CS15	NILM	NILM	NILM	NILM	NILM	NILM	LSIL	NILM	HSIL	ASC-US
20s		CS16	NILM	NILM	NILM	ASC-US	LSIL	NILM	NILM	ASC-US	HSIL	SCC

**Table 5 cancers-14-01159-t005:** Interobserver variability: kappa.

Classification	Dx Report	16 Cytoscreeners	8 Cytoscreeners (≥10 Years of Exp.)	
	NILM	0.042 (slight)	0.755 (substantial)	
Subclass	Neoplastic	0.098 (slight)	0.500 (moderate)	
	All cases	0.364 (fair)	0.716 (substantial)	
	NILM	0.073 (slight)	0.815 (almost perfect)	
Binary	Neoplastic	1.000 (complete)	1.000 (complete)	
	All cases	0.568 (moderate)	0.861 (almost perfect)	

## Data Availability

The datasets used in this study are not publicly available due to specific institutional requirements governing privacy protection; however, they are available from the corresponding author and from the private clinical laboratory in Japan on reasonable request. Restrictions apply based on the data use agreement, which was made according to the Ethical Guidelines for Medical and Health Research Involving Human Subjects as set by the Japanese Ministry of Health, Labour, and Welfare.

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
