# Peer review of "A Deep Learning Model for Cervical Cancer Screening on Liquid-Based Cytology Specimens in Whole Slide Images"

_cancers, 2022, doi:10.3390/cancers14051159_

Round 1
Reviewer 1 Report
Dear madam of sir,
this interesting paper proposes a new model for semi-automated cytoscreening based on LBC ThinPrep Pap test (Hologic, 79 Inc.) slide glass specimens that get digitised via specialized scanning devices into whole-slide images (WSI). The authors state, that this model could reduce the working time of cytoscreeners as the model is able to inference between non-neoplastic and neoplastic specimen and furthermore is able to highly the potential suspected neoplastic region in the WSI and thereby further reducing the screening time.
After carefully reading the paper, I cannot clearly detect which benefit this particular model offers compared to existing systems and particular to recently published models.
I see, that semi-automation in cytoscreening might be helpful, efficient and cost-effective particularly for regions with limited resources and could help to reduce the high burden of cervical cancer deaths in developing countries.
Could you clearly point out which particular benefits cytoscreeners and cytopathologists would have using this model compared to existing and recently published models. On the one hand, with regard to the already available models and on the other hand with regard to the recently published papers you mention in line 69 to 71.
In the lines 296 to 298 you state that “In routine cervical cancer screening at clinical laboratories and hospitals, it is difficult to introduce a screening programme dependent on cervical smears due to poor human cytoscreener resources.” This problem exists particularly in regions with limited resources especially in the developing countries, which also drive the high incidences of cervical cancers deaths. Is your model suitable for these regions?
Author Response
Reviewer 1:
this interesting paper proposes a new model for semi-automated cytoscreening based on LBC ThinPrep Pap test (Hologic, 79 Inc.) slide glass specimens that get digitised via specialized scanning devices into whole-slide images (WSI). The authors state, that this model could reduce the working time of cytoscreeners as the model is able to inference between non-neoplastic and neoplastic specimen and furthermore is able to highly the potential suspected neoplastic region in the WSI and thereby further reducing the screening time.
After carefully reading the paper, I cannot clearly detect which benefit this particular model offers compared to existing systems and particular to recently published models.
I see, that semi-automation in cytoscreening might be helpful, efficient and cost-effective particularly for regions with limited resources and could help to reduce the high burden of cervical cancer deaths in developing countries.
Could you clearly point out which particular benefits cytoscreeners and cytopathologists would have using this model compared to existing and recently published models. On the one hand, with regard to the already available models and on the other hand with regard to the recently published papers you mention in line 69 to 71.
Response: Unfortunately, none of the recently published models are publicly available, so we are not able to make use of them. Our models have also been trained with a different training methodology that involved doing less amount of work to prepare the training set. The trained models then offer both WSI level classification (NILM vs. neoplastic) and tile level prediction by heatmap, which are important for screening and are helpful for flexible operation for the person (or facility) checking the screening.
In the lines 296 to 298 you state that “In routine cervical cancer screening at clinical laboratories and hospitals, it is difficult to introduce a screening programme dependent on cervical smears due to poor human cytoscreener resources.” This problem exists particularly in regions with limited resources especially in the developing countries, which also drive the high incidences of cervical cancers deaths. Is your model suitable for these regions?
Response: Yes we do hope that such models would be particularly in regions with limited resources especially in the developing countries; however, we need to make sure that enough validation studies are conducted to ensure that the models can provide good performance for a variety of cases.
Reviewer 2 Report
The paper by Kanavati et al. entitled "A deep learning model for cervical cancer screening on liquid-based cytology specimens in whole slide images" reports the analysis of a global series of 1,605 cervical liquid-based cytology specimens using a deep learning model for the automated detection of cell abnormalities. The authors report a good performance of their model, with ROC AUCs variing from 0.89 to 0.96 according to the set of specimens analyzed. On the whole the work is original and the results promising. However, in my view, at this stage, this is only a methodological pilot study and further analyses are required to assess its potential clinical use.
The introduction underlines the high rate of "inconclusive results " (up to 30.8%) reported in literature for the automated analysis of cytological specimens using other devices. In the present work, no data are provided on this important point. It is stated (2.1) that "cases that had poor scanned quality were excluded: how many? or that (2.2) "cytoscreeners reviewed all the cases and the ones where they had a disagreement on where removed" (how many cases ? Were not ambiguous cases for which an automated analysis could be useful ?
Other remarks: (I) the Result paragraph is not easy to read. It begins by a general comment that forwards directly the reader to the tables without further explanations to understand the significance of the data presented in these tables. (II) The discussion starts with a long comment on the difference between conventional cytology VS liquid-based cytology, which is not relevant with the present topic. The issues raised in the introduction are not discussed.
On the whole, the paper describes a promising approach for the automated screening of liquid-based cervical cytological specimens. This is a methodological pilot study. The paper would be improved if the main objective of the study was clearly stated in the introduction and the results, regarding this objective, more clearly exposed and commented. In the discussion, perspectives on the potential clinical use of this approach could be mentionned as well as a possible strategy to reach this validation.
Author Response
Reviewer 2:
The paper by Kanavati et al. entitled "A deep learning model for cervical cancer screening on liquid-based cytology specimens in whole slide images" reports the analysis of a global series of 1,605 cervical liquid-based cytology specimens using a deep learning model for the automated detection of cell abnormalities. The authors report a good performance of their model, with ROC AUCs variing from 0.89 to 0.96 according to the set of specimens analyzed. On the whole the work is original and the results promising. However, in my view, at this stage, this is only a methodological pilot study and further analyses are required to assess its potential clinical use.
The introduction underlines the high rate of "inconclusive results " (up to 30.8%) reported in literature for the automated analysis of cytological specimens using other devices. In the present work, no data are provided on this important point. It is stated (2.1) that "cases that had poor scanned quality were excluded: how many? or that (2.2) "cytoscreeners reviewed all the cases and the ones where they had a disagreement on where removed" (how many cases ? Were not ambiguous cases for which an automated analysis could be useful ?
Response: We have added the number of poor scanned quality cases (n=32), and we have updated Table 1 with the number of cases that had disagreement.
The problem of the ambiguous cases is that for purposes of evaluating the model, it is difficult to know whether the model is making a correct or incorrect prediction if the cytoscreeners themselves have a disagreement on the diagnosis. They are potentially helpful in that the model could still bring them to the attention of the cytoscreeners for them to perform further action or consult with other cytoscreeners.
Other remarks:
(I) the Result paragraph is not easy to read. It begins by a general comment that forwards directly the reader to the tables without further explanations to understand the significance of the data presented in these tables.
Response: We have added a general summary of the overall results; however, we leave the discussion of the results to the discussion section.
(II) The discussion starts with a long comment on the difference between conventional cytology VS liquid-based cytology, which is not relevant with the present topic. The issues raised in the introduction are not discussed.
Response: We have moved that paragraph to the introduction.
On the whole, the paper describes a promising approach for the automated screening of liquid-based cervical cytological specimens. This is a methodological pilot study. The paper would be improved if the main objective of the study was clearly stated in the introduction and the results, regarding this objective, more clearly exposed and commented. In the discussion, perspectives on the potential clinical use of this approach could be mentionned as well as a possible strategy to reach this validation.
Response: We have added a mention that this is a pilot study in the introduction and discussion. And we have reworded the goal of the study in the abstract.